# Revisiting Dominance Pruning in Decoupled Search

**Daniel Gnad**

Saarland University
Saarland Informatics Campus
Saarbrücken, Germany
gnad@cs.uni-saarland.de

## Abstract

In classical planning as heuristic search, duplicate state pruning is a standard method to avoid unnecessarily handling the same state multiple times. In decoupled search, similar to symbolic search approaches, search nodes, called *decoupled states*, do not correspond to individual states, but to entire *sets* of states. As a result, duplicate state pruning cannot be applied in a straightforward manner. Instead, *dominance pruning* is employed, taking into account the state sets. We observe that the time required for dominance checking dominates the overall runtime, and propose two ways of tackling this issue. Our main contribution (1) is a stronger variant of dominance checking for optimal planning, where efficiency and pruning power are most crucial. The new variant greatly improves the latter, without incurring a computational overhead. Furthermore, (2) we develop and implement three methods that make the dominance check more efficient: exact duplicate checking, which, albeit resulting in weaker pruning, can pay off due to the use of hashing; avoiding the dominance check when leaf state spaces are invertible; and exploiting the transitivity of the dominance relation to only check against the relevant subset of visited decoupled states. We show empirically that all our improvements are indeed beneficial across many standard benchmark domains.

## Introduction

In classical planning, the most popular approach to solve planning tasks is heuristic search in the explicit state space (Bonet and Geffner 1999). Heuristic search, however, suffers from the state explosion problem that arises from the fact that the size of the state space of a task is exponential in the size of its description. Many methods have been introduced to tackle this explosion, such as partial-order reduction (Valmari 1989; Godefroid and Wolper 1991; Edelkamp, Leue, and Lluch-Lafuente 2004; Alkhazraji et al. 2012; Wehrle et al. 2013; Wehrle and Helmert 2014), symmetry breaking (Starke 1991; Pochter, Zohar, and Rosenschein 2011; Domshlak, Katz, and Shleyfman 2012), dominance pruning (Hall et al. 2013; Torralba and Hoffmann 2015; Torralba 2017), or symbolic representations (Bryant 1986; Edelkamp and Helmert 1999; Torralba et al. 2017). In this work, we look into a recent addition to this set of techniques, namely star-topology decoupled state space search, or decoupled search for short (Gnad and Hoffmann 2018).

Decoupled search is a form of factored planning (Amir and Engelhardt 2003; Brafman and Domshlak 2006, 2008; Fabre et al. 2010) that partitions the variables of a planning task into components such that the causal dependencies between the components form a star topology. The *center C* of this topology can interact arbitrarily with the other components, the *leaves* $\mathcal{L} = \{L_1, \ldots, L_n\}$, while any interaction between leaves has to involve the center, too. A *decoupled state* $s^{\mathcal{F}}$ corresponds to a single *center state*, an assignment to $C$, and a non-empty set of *leaf states* (assignments to an $L_i$) for each $L_i$. The *member states* of $s^{\mathcal{F}}$, i.e., the set of explicit states it represents, result from all combinations of leaf states across leaf factors, sharing the same center state. Thereby, a decoupled state represents exponentially many explicit states, leading to a reduction in search effort. Prior work has shown that there exist scalable families of planning tasks where this reduction is exponentially larger for decoupled search than it is for other state-space-reduction methods like partial-order reduction (Gnad, Hoffmann, and Wehrle 2019), symmetry breaking (Gnad et al. 2017), symbolic representations (Gnad and Hoffmann 2018), and Petri-net unfolding (Gnad and Hoffmann 2019).

Since decoupled states $s^{\mathcal{F}}$, correspond to *sets of states*, namely the member states of $s^{\mathcal{F}}$, the standard concept of duplicate elimination, ignoring a state that has already been visited (on a cheaper path) to avoid repeated work, cannot be applied so easily. More importantly, it is not as effective as in explicit state search, because two decoupled states are only equal if the entire sets of member states they represent are equal. Therefore, prior work on decoupled search only considered dominance pruning (Torralba et al. 2016; Gnad and Hoffmann 2018), where a decoupled state $s_1^{\mathcal{F}}$ that contains the member states $S_1$ is *dominated by* a decoupled state $s_2^{\mathcal{F}}$ that represents the set of states $S_2$ if $S_1 \subseteq S_2$, respecting the so-called *pricing function* of $s_1^{\mathcal{F}}$ and $s_2^{\mathcal{F}}$ in case of optimal planning. Then, if each member state of $s_2^{\mathcal{F}}$ is reached in $s_1^{\mathcal{F}}$ with at most the price it has in $s_2^{\mathcal{F}}$, we can safely prune $s_1^{\mathcal{F}}$, like duplicate states in explicit state search.

Initiating this work was the observation that on average around 60% of the overall runtime of decoupled search for optimal planning is spent on dominance checking (on the benchmarks from our evaluation that are solved in $\geq 0.1s$). Thus, we take a closer look at (1) algorithmic improvements that lead to an increased pruning power for optimal planning,

and (2) ways to make the dominance check more efficient in general. Regarding (1), we introduce two new extensions to the dominance check. First, we take into account not only the pricing function, but incorporate the $g$-value of $A^*$ in the check. Second, we propose a decoupled-state transformation that moves cost from the pricing function into the $g$-value. Both make the dominance check more informed, without introducing a computational overhead. For (2), we experiment with an implementation of exact duplicate checking, which, albeit resulting in weaker pruning, can be beneficial runtime-wise due to the use of hashing; we identify cases where leaves can be skipped in the check, namely if their *leaf state space* is invertible; and, exploiting the transitivity of the dominance relation, we only check against the non-dominated subset of visited decoupled states.

In our experimental evaluation, we see that the improvements as of (2) indeed have a (mostly mild) positive impact on the runtime. The stronger pruning variants from (1) lead to a substantial reduction in search effort, which translates to a strong runtime advantage.

## Background

We consider a classical planning framework with finite-domain state variables (Bäckström and Nebel 1995; Helmert 2006). In this framework a *planning task* is a tuple $\Pi = \langle \mathcal{V}, \mathcal{A}, I, G \rangle$, where $\mathcal{V}$ is a finite set of *variables*, each variable $v \in \mathcal{V}$ is associated with a finite domain $\mathcal{D}(v)$. $\mathcal{A}$ is a finite set of *actions*, each $a \in \mathcal{A}$ being a triple $\langle \mathsf{pre}(a), \mathsf{eff}(a), \mathsf{cost}(a) \rangle$ of *precondition*, *effect*, and *cost*. The preconditions $\mathsf{pre}(a)$ and effects $\mathsf{eff}(a)$ are partial assignments to $\mathcal{V}$, and the cost is a non-negative real number $\mathsf{cost}(a) \in \mathbb{R}^{0+}$. A *state* is a complete assignment to $\mathcal{V}$, $I$ is the *initial state*, and the *goal* $G$ is a partial assignment to $\mathcal{V}$. For a partial assignment $p$, we denote by $vars(p) \subseteq \mathcal{V}$ the subset of variables on which $p$ is defined. For $V' \subseteq \mathcal{V}$, by $p[V']$ we denote the restriction of $p$ onto $V' \cap vars(p)$, i. e., the assignment to $V'$ made by $p$. We identify (partial) variable assignments with sets of variable/value pairs.

An action $a$ is *applicable* in state $s$ if $\mathsf{pre}(a) \subseteq s$. Applying $a$ in a (partial) state $s$ changes the value of all $v \in vars(\mathsf{eff}(a)) \cap vars(s)$ to $\mathsf{eff}(a)[v]$, and leaves $s$ unchanged elsewhere. The outcome state is denoted $s[\![a]\!]$. A *plan* for $\Pi$ is an action sequence $\pi$ iteratively applicable in $I$, and resulting in a state $s_G$ where $G \subseteq s_G$. A plan $\pi$ is *optimal* if the summed-up cost of its actions, denoted $\mathsf{cost}(\pi)$, is minimal among all plans for $\Pi$.

During an $A^*$ search, we denote by $g(s)$ the minimum cost of a path on which a state $s$ was reached from $I$. Note that the $g$-value of a state can get reduced during the search, in case a cheaper path from $I$ to $s$ is generated.

### Decoupled Search

Decoupled search is a technique developed to avoid the combinatorial explosion of having to enumerate all possible variable assignments of causally independent parts of a planning task. It does so by partitioning the state variables into a *factoring* $\mathcal{F}$, whose elements are called *factors*. By imposing a structural requirement on the interaction between these factors, namely a *star topology*, decoupled search can efficiently handle cross-factor dependencies. A *star factoring* is one that has a *center* $C \in \mathcal{F}$ that interacts arbitrarily with the other factors $L \in \mathcal{L} := \mathcal{F} \setminus C$, called *leaves*, but where the only interaction between leaves is via the center.

Actions affecting $C$, i. e., with an effect on a variable in $C$, are called *center actions*, denoted $\mathcal{A}^C$, and those affecting a leaf are called *leaf actions*, denoted $\mathcal{A}^{\mathcal{L}}$. The actions that affect a particular leaf $L \in \mathcal{L}$ are denoted $\mathcal{A}^L$.[1] A sequence of center actions applicable in $I$ in the projection onto $C$ is a *center path*, a sequence of leaf actions affecting $L$, applicable in $I$ in the projection onto $L$, is a *leaf path*. A complete assignment to $C$, respectively an $L \in \mathcal{L}$, is called a *center state*, respectively *leaf state*. The set of all leaf states is denoted $S^{\mathcal{L}}$, and that of a particular leaf $L$ is denoted $S^L$.

A *decoupled state* $s^{\mathcal{F}}$ is a pair $\langle \mathsf{center}(s^{\mathcal{F}}), \mathsf{prices}(s^{\mathcal{F}}) \rangle$, where $\mathsf{center}(s^{\mathcal{F}})$ is a center state, and $\mathsf{prices}(s^{\mathcal{F}}) : S^{\mathcal{L}} \mapsto \mathbb{R}^{0+} \cup \{\infty\}$ is the *pricing function*, that assigns every leaf state a non-negative price. The pricing function is maintained during decoupled search in a way so that the price of a leaf state $s^L$ is the cost of a cheapest leaf path that ends in $s^L$ and that is *compliant*, i. e., that can be scheduled alongside the center path executed up to $s^{\mathcal{F}}$. By $S^{\mathcal{F}}$ we denote the set of all decoupled states. We say that a decoupled state $s^{\mathcal{F}}$ *satisfies* a condition $p$, denoted $s^{\mathcal{F}} \models p$, iff (i) $p[C] \subseteq \mathsf{center}(s^{\mathcal{F}})$ and (ii) for every $L \in \mathcal{L}$ there exists an $s^L \in S^L$ s.t. $p[L] \subseteq s^L$ and $\mathsf{prices}(s^{\mathcal{F}})[s^L] < \infty$. We define the set of leaf actions *enabled* by a center state $s^C$ as $\mathcal{A}^{\mathcal{L}}|_{s^C} := \{a^L \mid a^L \in \mathcal{A}^{\mathcal{L}} \wedge \mathsf{pre}(a^L)[C] \subseteq s^C\}$.

The *initial decoupled state* $s_0^{\mathcal{F}}$ is defined as $s_0^{\mathcal{F}} := \langle \mathsf{center}(s_0^{\mathcal{F}}), \mathsf{prices}(s_0^{\mathcal{F}}) \rangle$, where $\mathsf{center}(s_0^{\mathcal{F}}) = I[C]$. Its pricing function is given, for each $L \in \mathcal{L}$, as $\mathsf{prices}(s_0^{\mathcal{F}})[s_0^L] = 0$, where $s_0^L = I[L]$; and elsewhere as $\mathsf{prices}(s_0^{\mathcal{F}})[s^L] = c_{s_0^{\mathcal{F}}}(s_0^L, s^L)$, where $c_{s_0^{\mathcal{F}}}(s_0^L, s^L)$ is the cost of a cheapest path of $\mathcal{A}^L|_{\mathsf{center}(s_0^{\mathcal{F}})} \setminus \mathcal{A}^C$ actions from $s_0^L$ to $s^L$. If no such path exists $c_{s_0^{\mathcal{F}}}(s_0^L, s^L) = \infty$. The set of *decoupled goal states* is $S_G^{\mathcal{F}} := \{s_G^{\mathcal{F}} \mid s_G^{\mathcal{F}} \models G\}$.

Decoupled-state transitions are induced only by center actions, where a center action $a^C$ is *applicable* in a decoupled state $s^{\mathcal{F}}$ if $s^{\mathcal{F}} \models \mathsf{pre}(a^C)$. By $S_{a^C}^L(s^{\mathcal{F}})$ we define the set of leaf states of $L$ in $s^{\mathcal{F}}$ that *comply* with the leaf precondition of $a^C$, i. e., $S_{a^C}^L(s^{\mathcal{F}}) := \{s^L \mid \mathsf{pre}(a^C)[L] \subseteq s^L \wedge \mathsf{prices}(s^{\mathcal{F}})[s^L] < \infty\}$. Applying $a^C$ to $s^{\mathcal{F}}$ results in the decoupled state $t^{\mathcal{F}} = s^{\mathcal{F}}[\![a^C]\!]$ as follows: $\mathsf{center}(t^{\mathcal{F}}) = \mathsf{center}(s^{\mathcal{F}})[\![a^C]\!]$, and $\mathsf{prices}(t^{\mathcal{F}})[t^L] = \min_{s^L \in S_{a^C}^L(s^{\mathcal{F}})}(\mathsf{prices}(s^{\mathcal{F}})[s^L] + c_{t^{\mathcal{F}}}(u^L, t^L))$, where $s^L[\![a^C]\!] = u^L$.

By $\pi^C(s^{\mathcal{F}})$ we denote the center path that starts in $s_0^{\mathcal{F}}$ and ends in $s^{\mathcal{F}}$. Accordingly, we define the $g$-value of $s^{\mathcal{F}}$ as $g(s^{\mathcal{F}}) = \mathsf{cost}(\pi^C(s^{\mathcal{F}}))$, the cost of its center path.

A decoupled state $s^{\mathcal{F}}$ represents a set of explicit states, which takes the form of a *hypercube* whose dimensions are the leaf factors $\mathcal{L}$. Hypercubes are defined as follows:

---

[1] We remark that actions can be center and leaf action at the same time, so possibly $\mathcal{A}^{\mathcal{L}} \cap \mathcal{A}^C \neq \emptyset$.

**Definition 1 (Hypercube)** *Let $\Pi$ be a planning task, and $\mathcal{F}$ a star factoring. Then a state $s$ of $\Pi$ is a* member state *of a decoupled state $s^{\mathcal{F}}$, if $s[C] = \text{center}(s^{\mathcal{F}})$ and, for all leaves $L \in \mathcal{L}$, $\text{prices}(s^{\mathcal{F}})[s[L]] < \infty$. We say that $s$ has* price $\text{price}(s^{\mathcal{F}}, s)$ *in $s^{\mathcal{F}}$, where $\text{price}(s^{\mathcal{F}}, s) := \sum_{L \in \mathcal{L}} \text{prices}(s^{\mathcal{F}})[s[L]]$. The* hypercube *of $s^{\mathcal{F}}$, denoted $[s^{\mathcal{F}}]$, is the set of all member states of $s^{\mathcal{F}}$.*

The hypercube of $s^{\mathcal{F}}$ captures both, the reachability and the prices of all member states $s$ of $s^{\mathcal{F}}$. For every member state $s$ of a decoupled state $s^{\mathcal{F}}$, we can construct the *global plan*, i.e., the sequence of actions that starts in $I$ and ends in $s$ by augmenting $\pi^C(s^{\mathcal{F}})$ with cheapest-compliant leaf paths, i.e., leaf action sequences that lead to the pricing function of $s^{\mathcal{F}}$. The cost of member states in a hypercube only takes into account the cost of the leaf actions, since center action costs are not included in the pricing function. The cost of a plan reaching a member state $s$ of $s^{\mathcal{F}}$ from $I$ can be computed as follows: $\text{cost}(s^{\mathcal{F}}, s) = g(s^{\mathcal{F}}) + \text{price}(s^{\mathcal{F}}, s)$.

## Dominance Pruning for Decoupled Search

Prior work on decoupled search has only considered dominance pruning instead of exact duplicate checking (Torralba et al. 2016; Gnad and Hoffmann 2018). With dominance pruning, instead of duplicate states, the search prunes decoupled states that are *dominated* by an already visited decoupled state (with lower $g$-value). We formally define the dominance relation $\preceq_B \subseteq S^{\mathcal{F}} \times S^{\mathcal{F}}$ over decoupled states as $(t^{\mathcal{F}}, s^{\mathcal{F}}) \in \preceq_B$ iff (1) $[t^{\mathcal{F}}] \subseteq [s^{\mathcal{F}}]$ and (2) for all $s^L \in S^{\mathcal{L}}$ : $\text{prices}(s^{\mathcal{F}})[s^L] \leq \text{prices}(t^{\mathcal{F}})[s^L]$. Instead of $(t^{\mathcal{F}}, s^{\mathcal{F}}) \in \preceq_B$, we often write $t^{\mathcal{F}} \preceq_B s^{\mathcal{F}}$ to denote that $s^{\mathcal{F}}$ *dominates* $t^{\mathcal{F}}$. Note that (2) is only required for optimal planning. In satisficing planning we can simply set the price of all reached leaf states to 0, ignoring the leaf action costs completely. In practice these checks are performed by first comparing the center states $\text{center}(s^{\mathcal{F}}) = \text{center}(t^{\mathcal{F}})$ via hashing, followed by a component-wise comparison of the prices of reached leaf states.

## Exact Duplicate Checking

In explicit state search, duplicate checking is performed to avoid unnecessary repeated handling of the same state. This can be implemented efficiently by means of hashing functions: if a state is re-visited during search – and, in case of optimal planning using $A^*$, the path on which it is reached is not cheaper than its current $g$-value – the new state can be pruned safely. In this section, we will look into exact duplicate checking for decoupled search, showing how an efficient hashing can be implemented.

Formally, we define the *duplicate state relation* over decoupled states $\preceq_D \subseteq S^{\mathcal{F}} \times S^{\mathcal{F}}$ as the identity relation where $(t^{\mathcal{F}}, s^{\mathcal{F}}) \in \preceq_D$ iff $s^{\mathcal{F}} = t^{\mathcal{F}}$. Like in explicit state search, a decoupled state $t^{\mathcal{F}}$ can safely be pruned if there exists an already visited state $s^{\mathcal{F}}$ where $g(s^{\mathcal{F}}) \leq g(t^{\mathcal{F}})$ and $t^{\mathcal{F}} \preceq_D s^{\mathcal{F}}$.

In decoupled search, a search node, i.e., a decoupled state $s^{\mathcal{F}}$, does not represent a single state, but a set of states, namely its hypercube $[s^{\mathcal{F}}]$. As a consequence, duplicate checking is less effective. This is because the chances of

finding a decoupled state with the exact same hypercube (including leaf state prices) are smaller than finding a duplicate in explicit state search. Importantly, care must be taken when hashing decoupled states, to properly take into account both reachability *and* prices of leaf states.

In order to hash decoupled states, we need a canonical form that provides a unique representation of a decoupled state. We achieve this by, prior to the search, constructing *all* reachable leaf states $s^L$ for each leaf $L$, over-approximating reachability by projecting the task onto $L$. This ignores all interaction between the center and the leaf, assuming that action preconditions on $\mathcal{V} \setminus L$ are always achieved. The resulting transition systems are called the *leaf state spaces* for every leaf $L \in \mathcal{L}$. Given the leaf state spaces, we assign a unique ID to every leaf state, starting with 0, up to $|S_R^L| - 1$, where $S_R^L$ is the set of leaf states of $L$ that can be reached from $I[L]$ in the leaf state space of $L$.

With the leaf state IDs, we can efficiently store the pricing function of each leaf $L \in \mathcal{L}$ of a decoupled state $s^{\mathcal{F}}$ as an array $A$ of numbers, where $A[i]$ contains the price of the leaf state with ID $i$. To get a canonical representation of $s^{\mathcal{F}}$, and to keep the memory footprint of its pricing function as small as possible, we decide to limit the size of the array to just fit the highest ID of a leaf state with finite price. Implicitly, all leaf states with a higher ID are not reached in $s^{\mathcal{F}}$ and have cost $\infty$. This does incur a memory overhead, in the extreme case wasting $|S_R^L| - 1$ entries in the array, if only the leaf state with ID $|S_R^L| - 1$ is reached, so the entries for all other leaf states are $\infty$. However, leaf state spaces are mostly "well-behaved" in the sense that such pathologic behaviour does not usually occur.

In non-optimal planning, where, as previously noted, we do not require the actual leaf state prices, but only reachability information, we only keep a bitvector $A$ for each leaf $L \in \mathcal{L}$, indicating whether the leaf state with ID $i$ is reached if $A[i] = \top$.

The unique IDs and the maintenance of the pricing function as standard arrays allow the use of hashing functions, where two decoupled states can only be equal if the hashes of their center state, and for each leaf factor, the hashes of the representation of the pricing functions match.

## Improved Dominance for Optimal Planning

In this section, we introduce two improvements over the basic dominance relation $\preceq_B$ for optimal planning. The first one incorporates the $g$-value of decoupled states into the dominance check when comparing the leaf-state prices. This increases the potential for pruning, allowing to prune states that have a lower $g$-value. The second technique is a decoupled-state transformation that moves part of the leaf-state prices into the $g$-value of a decoupled state, enhancing search guidance by fully accounting for costs that have to be spent to reach the cheapest member state.

### Incorporating the $g$-value in Dominance Checking

In optimal planning, a decoupled state $t^{\mathcal{F}}$ can only be pruned with $\preceq_B$ if there exists an already visited state $s^{\mathcal{F}}$ with *lower g-value* that dominates it. Doing the dominance check in

$$\boxed{\begin{array}{ll} t^{\mathcal{F}} : g(t^{\mathcal{F}}) = 10 \\ s_1^L \to 1 \quad s_1^{L'} \to x \\ s_2^L \to 1 \quad s_2^{L'} \to x \end{array}} \begin{array}{c} \not\preceq_B \\ \preceq_G \end{array} \boxed{\begin{array}{ll} s^{\mathcal{F}} : g(s^{\mathcal{F}}) = 5 \\ s_1^L \to 6 \quad s_1^{L'} \to x \\ s_2^L \to 6 \quad s_2^{L'} \to x \end{array}}$$

Figure 1: Illustrating example where $s^{\mathcal{F}}$ can only be pruned when using the new dominance relation $\preceq_G$.

$$\boxed{\begin{array}{ll} t^{\mathcal{F}} : g(t^{\mathcal{F}}) = 10 \\ s_1^L \to 1 \quad s_1^{L'} \to 3 \\ s_2^L \to 1 \quad s_2^{L'} \to 3 \end{array}} \begin{array}{c} \not\preceq_B \\ \preceq_G \end{array} \boxed{\begin{array}{ll} s^{\mathcal{F}} : g(s^{\mathcal{F}}) = 5 \\ s_1^L \to 6 \quad s_1^{L'} \to 1 \\ s_2^L \to 6 \quad s_2^{L'} \to 1 \end{array}}$$

Figure 2: Example where $t^{\mathcal{F}}$ has lower prices in leaf $L$, higher prices in $L'$, and $\preceq_G$ detects that $s^{\mathcal{F}}$ dominates $t^{\mathcal{F}}$.

this way, however, separately considers $g$-values and pricing function, where these in fact can be combined. We next show that the $g$-value difference of two decoupled states can be traded against differences in the pricing function. To see this, recall the definition of the cost of a member state $s$ of a decoupled state $s^{\mathcal{F}}$:

$$\begin{aligned} \mathsf{cost}(s^{\mathcal{F}}, s) &= g(s^{\mathcal{F}}) + \mathsf{price}(s^{\mathcal{F}}, s) \\ &= g(s^{\mathcal{F}}) + \sum_{L \in \mathcal{L}} \mathsf{prices}(s^{\mathcal{F}})[s[L]] \end{aligned}$$

Instead of only comparing the pricing function to visited states with lower $g$-value, we can directly compare the costs of the member states to all visited states, independent of their $g$-values. Then, a new decoupled state $t^{\mathcal{F}}$ can be pruned if there exists a visited state $s^{\mathcal{F}}$ s.t. all member states $s$ of $t^{\mathcal{F}}$ have lower cost in $s^{\mathcal{F}}$: $\forall s \in [t^{\mathcal{F}}] : \mathsf{cost}(s^{\mathcal{F}}, s) \le \mathsf{cost}(t^{\mathcal{F}}, s)$. If all member states $s$ of a decoupled state $t^{\mathcal{F}}$ are contained with lower cost in an already visited decoupled state $s^{\mathcal{F}}$, then analogously to pruning duplicate states with higher $g$-value in explicit search, we can safely prune $t^{\mathcal{F}}$.

Consider the example in Figure 1. Each box represents a decoupled state, and an arrow $s_i^L \to 6$ indicates that in a state $s^{\mathcal{F}}$ we have $\mathsf{prices}(s^{\mathcal{F}})[s_i^L] = 6$. Say $s^{\mathcal{F}}$ is visited and $t^{\mathcal{F}}$ is a new state, where $g(t^{\mathcal{F}}) = 10$ and $g(s^{\mathcal{F}}) = 5$. Further, the prices in all but one leaf factor $L$ of both states are identical. In leaf $L$, we have $\mathsf{prices}(s^{\mathcal{F}})[s^L] = \mathsf{prices}(t^{\mathcal{F}})[s^L] + 5$, so all leaf states of $L$ in $t^{\mathcal{F}}$ are cheaper by a cost of 5, but $s^{\mathcal{F}}$ has a $g$-value that is by 5 lower than that of $t^{\mathcal{F}}$. With the dominance relation $\preceq_B$ from prior work, $t^{\mathcal{F}}$ cannot be pruned, because its prices are lower than the ones of $s^{\mathcal{F}}$. However, the cost of all its member states is equal to the cost of the states in $s^{\mathcal{F}}$, so it is actually safe to prune $t^{\mathcal{F}}$.

An important question is how to compute this efficiently, i.e., without explicitly enumerating the costs of all member states. We next show that, similar to $\preceq_B$, dominance can be checked component-wise by only considering, for each leaf $L$, the leaf state with the highest price difference.

Formally, we define the *g-value aware dominance rela-*

*tion* $\preceq_G \subseteq S^{\mathcal{F}} \times S^{\mathcal{F}}$ as follows:

$$\begin{aligned} (t^{\mathcal{F}}, s^{\mathcal{F}}) \in \preceq_G &\Leftrightarrow g(t^{\mathcal{F}}) - g(s^{\mathcal{F}}) \ge \\ &\sum_{L \in \mathcal{L}} max_{s^L \in S_R^L}(\mathsf{prices}(s^{\mathcal{F}})[s^L] - \mathsf{prices}(t^{\mathcal{F}})[s^L]), \\ &\text{where } S_R^L = \{s^L \in S^L \mid \mathsf{prices}(t^{\mathcal{F}})[s^L] < \infty\} \end{aligned}$$

If $t^{\mathcal{F}}$ has a higher $g$-value than $s^{\mathcal{F}}$, but has leaf states with a lower price, then the disadvantage in $g$-value can be traded against the advantage in leaf state prices. More concretely, it suffices to sum-up only the maximal price-difference of any leaf state over the leaves. Thereby, we essentially compare only the member state $s \in [t^{\mathcal{F}}]$ for which the price-advantage is maximal. This can be done component-wise, so is efficient to compute. Indeed, $\preceq_G$ detects that $t^{\mathcal{F}}$ in the above example is dominated and can be pruned.

**Theorem 1** *Let $s^{\mathcal{F}}$ and $t^{\mathcal{F}}$ be two decoupled states. Then $t^{\mathcal{F}} \preceq_G s^{\mathcal{F}}$ iff for all $s \in [t^{\mathcal{F}}] : \mathsf{cost}(t^{\mathcal{F}}, s) \ge \mathsf{cost}(s^{\mathcal{F}}, s)$.*

**Proof Sketch:** Let $s$ be the member state of $t^{\mathcal{F}}$ where $\mathsf{prices}(s^{\mathcal{F}})[s[L]] - \mathsf{prices}(t^{\mathcal{F}})[s[L]]$ is maximal for all $L \in \mathcal{L}$. If $\mathsf{prices}(s^{\mathcal{F}}, s) - \mathsf{prices}(t^{\mathcal{F}}, s) \le g(t^{\mathcal{F}}) - g(s^{\mathcal{F}})$, then this also holds for all other $s' \in [t^{\mathcal{F}}]$. With $\mathsf{cost}(t^{\mathcal{F}}, s') = g(t^{\mathcal{F}}) + \mathsf{prices}(t^{\mathcal{F}}, s')$ the claim follows. $\square$

The new relation $\preceq_G$ also tackles more subtle cases, where prices differ in several leaf factors. We then need to *distribute* the difference in $g$-values *across* the leaf factors, i.e., we cannot use the full difference for each factor. However, we can even trade lower prices in one leaf by higher prices in another, incorporating these different prices in the $g$-difference. Consider the example in Figure 2, which extends the previous example by a leaf factor $L'$ where $\mathsf{prices}(t^{\mathcal{F}})[s^{L'}] = \mathsf{prices}(s^{\mathcal{F}})[s^{L'}] + 2$ for all $s^{L'} \in S^{L'}$. We can then combine the price advantage of $+2$ in $L'$ for $s^{\mathcal{F}}$ with its $g$-advantage $+5$ to make up for a total price disadvantage of 7 in other leaves, where $t^{\mathcal{F}}$ might have lower prices:

$$\begin{aligned} g(t^{\mathcal{F}}) - g(s^{\mathcal{F}}) &= 5 \\ &\ge \sum_{L \in \mathcal{L}} max_{s^L \in S_R^L}(\mathsf{prices}(s^{\mathcal{F}})[s^L] - \mathsf{prices}(t^{\mathcal{F}})[s^L]) \\ &= (6-1) + (1-3) = 3 \implies t^{\mathcal{F}} \preceq_G s^{\mathcal{F}} \end{aligned}$$

In this case, given that $s^{\mathcal{F}}$ is visited before $t^{\mathcal{F}}$ during search, we can prune $t^{\mathcal{F}}$, although the prices of its leaf states are neither lower-equal, nor higher-equal than the prices of $t^{\mathcal{F}}$. There is even a difference of cost 2 left that could be used to trade higher prices of $s^{\mathcal{F}}$ in another leaf factor.

### g-Value Adaptation

We next introduce a canonical form which moves as much of the leaf-state prices into the $g$-value of a decoupled state as possible. Assume that, in a decoupled state $s^{\mathcal{F}}$, there exists a leaf $L$ such that all leaf states $s^L$ have a minimum non-zero price $p$, so $\forall s^L \in S^L : \mathsf{prices}(s^{\mathcal{F}})[s^L] \ge p$. Then we can reduce the prices of all these leaf states by $p$ and increase $g(s^{\mathcal{F}})$ by $p$ without affecting the cost $\mathsf{cost}(s^{\mathcal{F}}, s)$

$$\boxed{\begin{array}{ll} s^{\mathcal{F}} : g(s^{\mathcal{F}}) = 5 \\ s_1^L \to 3 \quad s_1^{L'} \to 2 \\ s_2^L \to 1 \quad s_2^{L'} \to 3 \end{array}} \to \boxed{\begin{array}{ll} t^{\mathcal{F}} : g(t^{\mathcal{F}}) = 8 \\ s_1^L \to 2 \quad s_1^{L'} \to 0 \\ s_2^L \to 0 \quad s_2^{L'} \to 1 \end{array}}$$

Figure 3: A decoupled state $s^{\mathcal{F}}$ and its $g$-adapted variant $t^{\mathcal{F}}$.

of the member states $s$ of $s^{\mathcal{F}}$. Intuitively, the transformation moves the price that has to be spent to reach the cheapest member state of $s^{\mathcal{F}}$ into its $g$-value, reducing the price of all leaf states accordingly, so that in every leaf $L$ there exists at least one leaf state with price 0. See Figure 3 for an example of a decoupled state $s^{\mathcal{F}}$ and its canonical representative $t^{\mathcal{F}}$.

The main advantage of adapting the $g$-value of a decoupled state occurs when executing decoupled search using the $A^*$ algorithm. Here, on a cost layer $f$ the search usually prioritizes states with lower heuristic value. By moving cost into the $g$-value we achieve that the heuristic of a decoupled state (which takes into account the pricing function (Gnad and Hoffmann 2018)) can only get lower, aiding $A^*$ to focus on more promising states. A second important effect is that the part of the prices moved into the $g$-value will always be considered entirely by the search, whereas heuristics (in the extreme case blind search) might not be able to capture all the cost encoded in the pricing function.

Note that the $g$-value adaptation is independent of the new dominance relation $\preceq_G$. It can have a positive impact on the number of state expansions of $\preceq_G$, the base dominance check $\preceq_B$, and exact duplicate checking $\preceq_D$.

## Efficient Implementation

In this section, we propose two optimizations that aim at making the dominance check more efficient. First, we show that with *invertible* leaf state spaces the comparison of leaf reachability can be entirely avoided. Second, we show how to exploit the *transitivity* of the dominance relation to focus the checking on the relevant subset of decoupled states. Both optimizations do not affect the pruning behavior.

### Invertible Leaf State Spaces

Given the precomputed leaf state spaces as described in the previous section, it is straightforward to compute the connectivity of these graphs. In particular, we can efficiently check if a leaf state space is strongly connected when only considering transitions of leaf actions that do not affect, nor are preconditioned by, the center factor. Formally, we define the set of *no-center* actions of a leaf $L$ as $\mathcal{A}_{\neg C}^L := \{a^L \in \mathcal{A}^L \mid vars(\mathsf{pre}(a)) \cap C = \emptyset \wedge vars(\mathsf{eff}(a)) \cap C = \emptyset\}$.

Let $S_R^L$ be the set of $L$-states that is reachable from $I[L]$ in the projection onto $L$ using all actions $\mathcal{A}$. Let further $S_R^L|_{\mathcal{A}_{\neg C}^L}$ be the corresponding set using only the no-center actions $\mathcal{A}_{\neg C}^L$ of $L$. We say that $L$ is *leaf-invertible*, if $S_R^L = S_R^L|_{\mathcal{A}_{\neg C}^L}$, i.e., any $L$-state reachable from $I[L]$ can be reached using no-center actions, and the part of the leaf state space induced by $S_R^L$ and $\mathcal{A}_{\neg C}^L$ is strongly connected.

**Proposition 1** *Let $L$ be leaf-invertible and $S_R^L$ the set of $L$-states reachable from $I[L]$, then in every decoupled state $s^{\mathcal{F}}$*

*reachable from $s_0^{\mathcal{F}}$, the set of reached $L$-states in $s^{\mathcal{F}}$ is $S_R^L$.*

**Proof:** In $s_0^{\mathcal{F}}$, the claim trivially holds. Let $s^{\mathcal{F}}$ be a (not necessarily direct) successor of $s_0^{\mathcal{F}}$. The center action that generates $s^{\mathcal{F}}$ can possibly restrict the set of compliant leaf states $S_a^L$, and affect the remaining ones, resulting in a set of leaf states that is a subset of $S_R^L$. Since $S_R^L$ is strongly connected by $\mathcal{A}_{\neg C}^L$, all $L$-states of $S_R^L$ have a finite price in $s^{\mathcal{F}}$. $\square$

All decoupled states reached during search can only differ in the leaf-state prices for leaf-invertible factors, but will always have the same set of leaf states reached. Thus, at least for satisficing planning, these leaves do not need to be compared in the dominance check at all. For optimal planning, we still need to compare the prices, since these might differ.

Another minor optimization that can be performed with the leaf-invertibility information is successor generation during search. When computing the center actions that are applicable in a decoupled state, we usually need to check leaf preconditions by looking for a reached leaf state that enables an action. For leaf-invertible leaf factors, however, this check is no longer needed (even for optimal planning), because the set of reached leaf states remains constant. We precompute the set of applicable center actions, and skip the check for leaf preconditions on leaf-invertible factors.

### Transitivity of the Dominance Relation

In explicit state search, a duplicate state can be pruned if it has already been visited (with a lower $g$-value). This can be efficiently implemented using a hash table. In decoupled search with dominance pruning, the corresponding check needs to iterate over *all* previously visited states (with a lower $g$-value) that have the same center state, and compare the pricing function.

Instead of iterating over *all* visited decoupled states, though, we can exploit the transitivity of our dominance relations to focus on the relevant visited states, namely those that are not themselves dominated by another visited state.

**Proposition 2** *Let $V$ be the set of decoupled state already visited during search and let $t^{\mathcal{F}}$ a newly generated decoupled state. If there exist $s_1^{\mathcal{F}}, s_2^{\mathcal{F}} \in V$ such that $s_1^{\mathcal{F}} \preceq s_2^{\mathcal{F}}$, where $\preceq$ is a transitive relation over decoupled states, then $t^{\mathcal{F}} \not\preceq s_2^{\mathcal{F}}$ implies $t^{\mathcal{F}} \not\preceq s_1^{\mathcal{F}}$.*

Clearly, we do not need to check dominance of $t^{\mathcal{F}}$ against $s_1^{\mathcal{F}}$, but only need to compare $s_2^{\mathcal{F}}$ and $t^{\mathcal{F}}$ to see if $t^{\mathcal{F}}$ can be pruned. During search, we incrementally compute the set of "dominated visited states" as a side product of the dominance check. If a new state $t^{\mathcal{F}}$ dominates an existing state $s_1^{\mathcal{F}}$, then either there exists another visited state $s_3^{\mathcal{F}}$ that dominates $t^{\mathcal{F}}$, so it will be pruned, or there is no state yet that dominates $t^{\mathcal{F}}$. In both cases, $s_1^{\mathcal{F}}$ can be skipped in every future dominance check because there exists another visited state, either $s_3^{\mathcal{F}}$ or $t^{\mathcal{F}}$, that is visited and that dominates it.

## Experimental Evaluation

We implemented all proposed methods in the decoupled search planner by Gnad & Hoffmann (2018), which itself builds on the Fast Downward planning system (Helmert

Table 1:

| Domain | # | Blind Search Dominance Pruning $\preceq_B$ | $\preceq_B^{IT}$ | $\preceq_B^{gIT}$ | $\preceq_G^{IT}$ | $\preceq_G^{g}$ | $\preceq_G^{gIT}$ | Duplicate Checking $\preceq_D$ | $\preceq_D^{I}$ | $\preceq_D^{g}$ | $\preceq_D^{gI}$ | A* with $h^{\text{LM-cut}}$ Dominance Pruning $\preceq_B$ | $\preceq_B^{IT}$ | $\preceq_B^{gIT}$ | $\preceq_G^{IT}$ | $\preceq_G^{g}$ | $\preceq_G^{gIT}$ | Duplicate Checking $\preceq_D$ | $\preceq_D^{I}$ | $\preceq_D^{g}$ | $\preceq_D^{gI}$ |
|---|---|---|---|---|---|---|---|---|---|---|---|---|---|---|---|---|---|---|---|---|---|
| DataNet | 20 | **9** | **9** | 5 | **9** | **9** | **9** | 5 | 5 | 5 | 5 | **14** | **14** | 12 | **14** | **14** | **14** | 12 | 12 | 12 | 12 |
| Depots | 22 | 3 | 3 | 4 | 4 | 4 | 4 | 2 | 2 | 4 | 4 | 7 | 7 | 7 | 7 | 7 | 7 | 5 | 5 | 7 | 7 |
| Driverlog | 20 | **11** | **11** | **11** | **11** | **11** | **11** | 9 | 9 | 10 | 10 | 13 | 13 | 13 | 13 | 13 | 13 | 13 | 13 | 13 | 13 |
| Elevators | 30 | 6 | 6 | 9 | 12 | **16** | **16** | 0 | 0 | 10 | 10 | 10 | 11 | 22 | 13 | **23** | **23** | 0 | 0 | 22 | 22 |
| Floortile | 40 | **2** | **2** | **2** | **2** | **2** | **2** | 0 | 0 | 0 | 0 | **10** | **10** | **10** | **10** | **10** | **10** | 5 | 5 | 5 | 5 |
| Freecell | 42 | 0 | 0 | 0 | 0 | 0 | 0 | 0 | 0 | 2 | 2 | 1 | 1 | 2 | 2 | 2 | 2 | 1 | 1 | 2 | 2 |
| GED | 20 | 13 | 13 | **15** | **15** | **15** | **15** | 7 | 7 | **15** | **15** | 15 | 15 | 15 | 15 | 15 | 15 | 13 | 13 | 15 | 15 |
| Grid | 5 | 1 | 1 | 1 | 1 | 1 | 1 | 1 | 1 | 1 | 1 | 2 | 2 | 2 | 2 | 2 | 2 | 1 | 1 | 2 | 2 |
| Logistics | 63 | 24 | 25 | 25 | 26 | 25 | 26 | 22 | 22 | 24 | 24 | 34 | 34 | 35 | 34 | 36 | 36 | 30 | 30 | 34 | 34 |
| Miconic | 145 | 46 | **47** | **47** | **47** | 45 | **47** | 42 | 42 | 42 | 42 | 135 | 135 | 135 | 135 | 135 | 135 | 135 | 135 | 135 | 135 |
| NoMystery | 20 | **20** | **20** | **20** | **20** | 19 | **20** | 16 | 16 | 16 | 16 | **20** | **20** | **20** | **20** | **20** | **20** | 19 | 19 | 19 | 19 |
| OpenSt14 | 20 | 1 | **2** | 1 | **2** | **2** | 1 | **2** | **2** | **2** | **2** | 2 | 2 | 1 | 2 | 2 | 1 | 2 | 2 | 2 | 2 |
| PSR | 48 | **48** | **48** | 46 | **48** | **48** | **48** | 42 | 42 | 46 | 46 | 48 | 48 | 47 | 48 | 48 | 48 | 45 | 45 | 47 | 47 |
| Rovers | 40 | **7** | **7** | **7** | **7** | **7** | **7** | 6 | 6 | 7 | 7 | 8 | 8 | 8 | 8 | 8 | 8 | 8 | 8 | 8 | 8 |
| Satellite | 36 | 5 | 5 | 5 | 5 | 5 | 5 | 5 | 5 | 5 | 5 | 7 | 7 | 9 | 7 | 9 | 9 | 7 | 7 | 9 | 9 |
| Tidybot14 | 10 | 3 | 3 | 3 | 3 | 3 | 3 | 3 | 3 | 3 | 3 | **6** | **6** | **6** | **6** | **6** | **6** | 6 | 6 | 5 | 6 |
| Transport | 28 | 10 | 11 | 13 | 14 | **15** | **15** | 0 | 0 | 13 | 13 | 12 | 12 | **14** | 13 | **14** | **14** | 6 | 6 | **14** | **14** |
| Woodwork | 26 | 7 | 7 | 7 | 7 | 7 | 7 | 7 | 7 | 7 | 7 | 16 | 16 | 16 | 17 | 17 | 17 | 16 | 16 | 16 | 16 |
| Zenotravel | 20 | 8 | **9** | 8 | **9** | **9** | **9** | 6 | 6 | 6 | 7 | 12 | 12 | 12 | 13 | 13 | 13 | 8 | 8 | 11 | 11 |
| Others | 239 | 67 | 67 | 67 | 67 | 67 | 67 | 67 | 67 | 67 | 67 | 87 | 87 | 87 | 87 | 87 | 87 | 87 | 87 | 87 | 87 |
| $\sum$ | 894 | 291 | 296 | 296 | 309 | 310 | **313** | 242 | 242 | 285 | 286 | 459 | 460 | 473 | 466 | **481** | 480 | 419 | 419 | 465 | 466 |

Table 1: Coverage data for optimal planning with blind search and with A* using $h^{\text{LM-cut}}$. All configurations use the incident arcs factoring. Domains without difference in coverage are summarized in "Other". Best coverage is highlighted in **bold face**.

2006). We conducted our experiments using the lab python package (Seipp et al. 2017) on all benchmark domains of the International Planning Competition (IPC) from 1998-2018 in both the optimal and satisficing tracks. We also run decoupled search to prove planning tasks unsolvable, using the benchmarks of UIPC'16 and Hoffmann, Kissmann, and Torralba (2014). In all benchmark sets, we eliminated duplicate instances that appeared in several iterations. For optimal planning, we run blind search and A* with $h^{\text{LM-cut}}$ (Helmert and Domshlak 2009); in satisficing planning, we use greedy best-first search (GBFS) with the $h^{\text{FF}}$ heuristic without preferred operator pruning (Hoffmann and Nebel 2001); to prove unsolvability, we run A* with the $h^{\max}$ heuristic (Bonet and Geffner 2001). We use the common runtime/memory limits of 30min/4GiB. The code and experimental data of our evaluation are publicly available (https://doi.org/10.5281/zenodo.4061825).

Decoupled search needs a method that provides a factoring, i.e., that detects a star topology in the causal structure of the input planning task. We use two basic factoring methods, inverted-fork factorings (IF) – only for satisficing planning – as well as the incident arcs factoring (IA) as described in Gnad, Poser, and Hoffmann (2017). We expect IF factorings to nicely show the advantage of the more efficient handling of invertible leaf state spaces, since there are several domains that have such state spaces in this case, but not using IA. IA is the canonical choice since it is fast to compute and finds good decompositions in many domains.

We use the following naming convention for search configurations: we distinguish the three dominance relations $\preceq_B$, $\preceq_D$, and $\preceq_G$. We indicate the $g$-value adaptation, and the invertibility and transitivity optimizations by adding a superscript $g$, respectively $I$ and $T$ to the relation symbol, e.g. $\preceq_B^{gT}$ for a configuration that uses $\preceq_B$ and has the $g$-value adaptation and the transitivity optimization enabled.

Tables 1 and 2 show coverage data (number of instances solved) for the benchmarks where the factoring methods are able to detect a star factoring. For optimal planning, Table 1, we see that both $\preceq_G$ and the $g$-adaptation individually lead to an increase in coverage across several domains for both blind search and A* with $h^{\text{LM-cut}}$. There even seems to be a positive correlation, shown by the fact that the combination $\preceq_G^g$ outperforms both its components. We do not separately evaluate the invertibility optimization, since the only change in the successor generation does not influence coverage a lot. Compared to $\preceq_B$, $\preceq_B^I$ only differs in coverage by $+1$ in Openstacks for blind search. The main advantage of the $\preceq_x^{IT}$ configurations stems from the transitivity optimization.

The duplicate checking configurations without $g$-adaptation show a significant drop in coverage compared to $\preceq_B$, so although the checking is computationally more

| Unsolvability Domain | # | $\preceq_B^{IT}$ | $\preceq_D^{I}$ | Satisficing Domain | # | $\preceq_B^{IT}$ | $\preceq_D^{I}$ |
|---|---|---|---|---|---|---|---|
| Cavediving | 21 | **4** | 2 | | | | |
| Diagnosis | 20 | **14** | 13 | Floortile | 40 | **5** | 2 |
| OverNoMy | 24 | **13** | 12 | NoMyst | 20 | **19** | 16 |
| OverTPP | 30 | **15** | 14 | Rovers | 40 | **21** | 20 |
| Other | 182 | 88 | 88 | Other | 884 | 657 | 657 |
| $\sum$ | 277 | **134** | 129 | $\sum$ | 984 | **702** | 695 |

Table 2: Coverage data, like Table 1, for proving unsolvability and satisficing planning with preferred operators.

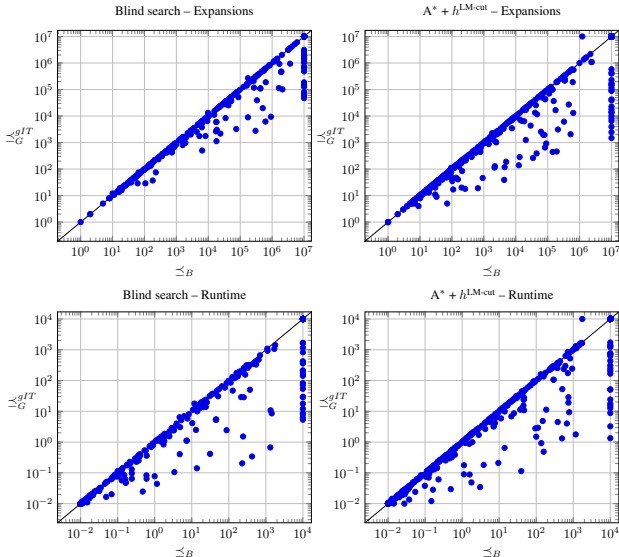

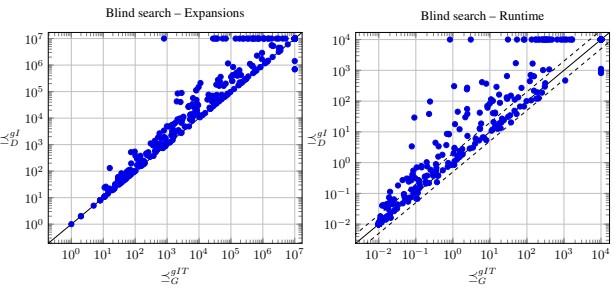

Figure 4: Scatter plots comparing runtime and number of state expansions for optimal planning.

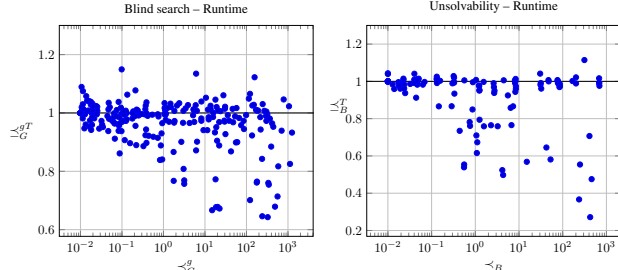

Figure 6: Improvement factors of $\preceq_y^{xT}$ over $\preceq_y^x$ showing the impact of the transitivity optimization in optimal planning and proving unsolvability.

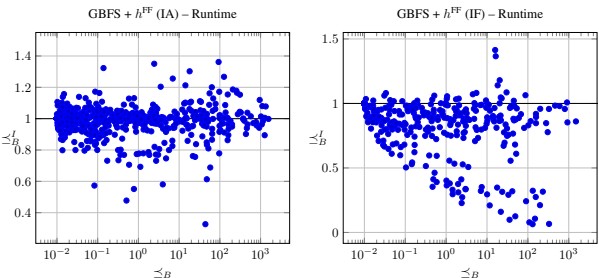

Figure 7: Like Figure 6, showing the impact of the invertibility optimization in satisficing planning with IA vs. IF.

Figure 5: Like Figure 4, comparing dominance pruning to duplicate checking.

efficient, this does not even out the weaker pruning power. When enabling the $g$-adaptation, total coverage gets almost back to the level of $\preceq_B$ for blind search, and increases by 7 instances for $h^{\text{LM-cut}}$. Compared to $\preceq_B^{gIT}$, though, in both search variants $\preceq_D^{gI}$ drops in coverage. Surprisingly, duplicate checking can solve two Freecell instances that the dominance pruning cannot solve with blind search.

Table 2 has coverage (number of instances solved, resp. proved unsolvable) results for proving unsolvability and satisficing planning. We focus on the difference between dominance pruning and duplicate checking, as the invertibility and transitivity optimizations do not have an impact on coverage. Duplicate checking performs worse in several domains, does not affect coverage across many other domains, but never increases coverage. So even outside optimal planning, with strong pruning being less crucial, its use does in general not pay off.

The scatter plots in Figure 4 and 5 shed further light on the per-instance runtime and search space size comparison between some optimal planning configurations. Figure 4 shows the number of expanded states in the top row and the

runtime in the bottom row. All configurations use the IA factoring and compare $\preceq_B$ to $\preceq_G^{gIT}$, with blind search in the left column and $h^{\text{LM-cut}}$ in the right column. The advantage of the more clever dominance check, the $g$-adaptation, and our runtime optimizations is obvious, saving up to several orders of magnitude for state expansions and runtime.

Figure 5 illustrates the effect of exact duplicate checking. The left plot shows the expected increase in search space size, due to the reduced pruning power. The right plot indicates that where the increase in search space size is small the more efficient computation indeed pays off runtime-wise. The two dashed lines highlight a difference of a factor of 2, which can often be gained with duplicate checking.

Figures 6 and 7 show the impact of the transitivity and invertibility optimizations. The plots show per-instance runtime improvement factors of configuration Y on the y-axis over configuration X on the x-axis, where a $y$-value of $a$ indicates that the runtime of Y is $a$ times the runtime of X (values below 1 are a speed-up). The transitivity optimization clearly has a positive impact on runtime, reducing it up to 40% in optimal planning and up to 70% for proving unsolvability. The invertibility optimization (Figure 7) does not show such a clear picture when using the IA factoring (left plot). With IF, though, it indeed nicely accelerates the dominance check, as the optimization is applicable more often.

## Conclusion

We have taken a closer look at the behavior and implementation details of dominance pruning in decoupled search. We

introduced exact duplicate checking, which, in spite of its weaker pruning, can improve search performance in practice due to higher computational efficiency under certain conditions. Furthermore, we developed two optimizations that make the dominance check more efficient to compute.

Our main contribution are two extensions of dominance pruning for optimal planning, that take the $g$-value of decoupled states into account. Both methods are highly beneficial and their combination significantly outperforms the baseline, improving the performance of decoupled search across many benchmark domains.

For future work, we think it is worthwhile to further investigate dominance pruning for decoupled search. A combination with the quantitative dominance pruning of Torralba (2017) could for example be interesting.

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
