# OpenReview forum: "An Analysis of Dominance Pruning in Decoupled Search"
_icaps-conference.org/ICAPS/2020/Workshop/HSDIP — HSDIP 2020_

### Official Review · AnonReviewer1 · 2020-03-31
**I think the paper is written very clearly, the theory seems sound and the experimental design makes sense.**

**Rating:** 9
**Confidence:** 4

**Review:**

Comments:

I think adding examples would make the paper more accessible.

For satisficing planning, you store a bitvector for each leaf. Couldn't you use the same "array for finite numbers" trick as for optimal planning and order the leaf reachable before the unreachable leaf states and then only store the number X of reachable leaf states? Then a leaf state with ID i is reachable if i < X.

The paper states that "checking dominance takes a large part in the overall search time". It would be interesting to see quantitative results for this.

I think it could be useful to note early in the paper that duplicate checking has less pruning power, but may be quicker to compute.

I recommend publishing the code and the results, preferably in an archival data store such as Zenodo.


Typos and style:

"much high++er++ g-value"

I'd remove the "literally" in both locations.

In the "Dominance Pruning" section there's a § which should probably be a $t$.

"store the pricing function of each leaf L \in \mathcal L ++for++ a decoupled state"

"leaf actions that do not affect--ed--"

I think the last part of the first sentence in "Incorporating the g-value in Dominance Checking" is missing a "\leq".

I'd drop "the standard" from the description of the runtime and memory limits, since there is no standard memory limit.

"Table--s-- 1"

"loose a lot" -> lose a lot, "looses a few" -> lose a few

---

### Official Review · AnonReviewer2 · 2020-04-01
**Full review already since I read the paper anyway**

**Rating:** 8
**Confidence:** 4

**Review:**

This paper analyzes pruning and duplicate detection in decoupled search. 1) It
investigates if (exact) duplicate detection instead of the usual dominance
pruning (because decoupled states represent sets of planning task states) is
feasible. To this end, the paper suggests implementation optimizations, and,
for satisficing planning, a method that does not need hashing at all if the
leaf state space is strongly connected. 2) The paper shows how g-values can be
incorporated into dominance pruning for optimal planning.

The paper is very well written and manages to explain decoupled search in a
very good way, although it is still quite technical and does not provide any
example. The paper is sound and provides an experimental evaluation. The topic
definitely fits HSDIP, therefore this is a clear accept.

Up until the experimental evaluation, I have only a few minor comments (see
below). While I like the theoretical contribution, I found the experimental
less convincing and I think that the results should be explained in more detail
and be sold to the reader in a much less pronounced way. Details follow.

The second paragraph, which lists the configurations that will be discussed,
was hard to match onto the previous section on pruning: it might help to
introduce the names (the abbreviations) already in the previous section. Also,
I didn't understand how DUP can be combined with the g-value improvement, as
this appears to be an optimization for dominance pruning. Furthermore, I
wondered if it would be worth to report the impact of the few implementation
optimizations - after all, you describe them in much detail. But then again,
given the marginal changes in coverage, I can imagine that without them, the
approaches are less competitive, or that they don't have a large impact. In
either case, I think this impact should be discussed at least, if not fully
reported.

The paragraph that discusses Table 1, to me, sounds like it is trying to
over-sell the results: "surprising results" (twice); "interesting" and "highly
domain-dependent" (we are talking about -3 coverage in two domains, compared to
-1/-2 in a few others!). I would suggest to leave such feelings/conclusions to
the reader :-) Also, I disagree that DUP does not loose "a lot" compared to
gIn: it is even below Base in total coverage, and people often argue that
already a change of only a few solved tasks more/less is already significant.

The text for Table 2 just states "the invertibility optimization does not
affect coverage". Can you elaborate please? Doesn't it apply often, because
leaf state spaces are not strongly connected? Are there other reasons?

The discussion of the Figures is quite hard to follow, because a) there are
many plots and b) the text never refers to specific plots, but only to the
global picture. Here, it would be really helpful to guide the reader towards
specific interesting plots, and to possibly include fewer plots as to highlight
the important observations. Also, what is "advanced dominance pruning" here?
Furthermore, I assume DUP is compared to iGn because indeed DUP includes iGn,
but then, again, I don't understand how. Also please introduce the
abbreviations used in the figures (such as IA, IF).

Finally, the paper left me unsure on what to conclude regarding the new
techniques: clearly, it always makes sense to use g-values for normal dominance
pruning. But any of the other techniques, I suppose, are not advisable to be
used? INV doesn't change anything, and exact duplicate detection is still to
expensive. Is that the right conclusion?

Other comments:
- involve to center -> involve the center
- high g-value -> higher g-value
- S_compl^L: this should depend on a^C (i.e., have a^C as subscript/
superscript), also the name compl could be related to "compliant" (which wasn't
obvious to me)
- there is a wrong symbol § in Section "Dominance Pruning"
- "subsection": somewhat technical, could just write "in the following", or
"section"
- how are leaf states stored/mapped to their IDs so as to benefit from the
ID-to-cost mapping during search?
- "paragraph" (twice): this actually means "subsection", so the above applies
- do not affected -> do not affect the center
- prices(t^F)[s^L] prices(s^F)[s]: missing <=
- equations: after the fourth =, the line should be prices(t^F)[s[L']]+2
instead of prices(S^F)[s[L']]+2
- price if t^F -> of
- Fast Downward: which version?
- will the code and data be published?

---

### Author Response · Authors · 2020-07-25
**Author response and changes in latest submission**

We thank both reviewers for the very constructive and positive feedback. We tried our best to address all your comments to improve the paper as much as possible. All of your questions are now clarified in the respective sections.

Regarding some specific questions:
- DUP vs. g-value improvement: this was indeed a mistake in the first version. What was meant is a combination of DUP and the g-value transformation newly introduced in the recent version of the paper.
- Regarding the conclusions: it is advisable to always use all improvements introduced in the paper, except for exact duplicate checking.
- We added some new plots in the introduction showing the ratio of the overall runtime spent in dominance checking.

Difference to previous version:
- Dominance checking time plots and short discussion thereof in introduction.
- The section "Incorporating the g-value in Dominance Checking" is almost entirely rewritten, introducing a new canonical form that moves minimum prices into the g-value of a decoupled state (at the end of the section). There are now also several examples.
- In the experiments, we addressed the comments to better explain the shown configurations and discuss the plots in more detail. We also show different plots, focusing on plots that clearly show the changes in search effort and runtime.

---

### Comment · AnonReviewer2 · 2020-08-04
**Further comments after reviewing the revised submission**

Thanks for improving this submission further. The paper is in a very good shape
now, and I only have a few comments, most of them still regarding the
experimental section, which is a lot clearer now but still has a few
inconsistencies.

Intro:

- I'm not sure I like the addition of the plots and the extensive discussion of
the portion of runtime used by dominance pruning that much, mainly because in
my view, stating that it can take a large part of runtime was enough as
motivation already. Anyway, if you decide to leave this in, I think that the
text discussing the plots should be much more abstract. Currently, it reads
like a discussion of an experimental evaluation, mentioning details of
algorithms, factorings and planning techniques all of which haven't been
mentioned/explained yet. (For example, "satisficing planning".)

If space was an issue for another submission, I would consider dropping this
again.

Background:

- What happens with actions that affect center and leaf variables? Are they
part of A^C or A^L?

- "A complete assignment to C is called a center state, an assignment to an L":
I assume also the latter is a complete assignment? I would either use complete
for both or for none.

- "The pricing function is given... where ... ; and elsewhere ...": use "if"
and "otherwise" instead of "where" and "elsewhere"; the sentence is hard to
parse.

Exact Duplicate Checking:

- "size of the array to just fit the price of the highest ID": I found this
confusing and had to read it many times. What confused me is the part "to fit
the price". I think it would be better to phrase it like "size of the array to
the highest ID of a leaf state with finite cost".

Incorporating the g-value:

- "The key observation": for what? -> the improved pruning

- The paragraph after recalling the definition of cost(s^F, s) feels
disconnected. I think starting it with "Therefore, a new decoupled state s^F
can be pruned" and making sure to highlight the difference to <=B (namely using
cost rather than price). Also "such a dominance relation" is not really clear
because the new dominance relation <=G will only be defined further down. Maybe
it is also worth considering first discussing Figure 2.

- "an arrow s_i^L -> 3": there is no such arrow in Figure 2

- "see for example Figure 3": I think it is to early to give this reference. I
would refer Figure 3 in the sentence starting with "Say, extending the above
example". Then it is also clear that "above example" refers to Figure 2 and not
3.

- The optimization that moves leaf costs to g-values (pInG) could be separated
into its own subsection. I don't really see that it would be connected to the
dominance pruning based on g-values.

- Figure 4 could be dropped if space was an issue.

Experimental Evaluation:

- "in Gnad et al.": -> by

- "both search types": I assume *planning* types (optimal/satisficing). For a
non-ICAPS audience, I would definitely make it clear that A* is used for
optimal and GBFS for satisficing planning.

- Some of the smaller optimization techniques still don't have their own name
and their own evaluation (or at least, it wasn't clear to me if they had): the
successor generation is always used with optimal planning (why?) but not with
satisficing planning? Why does DUP include INV for satisficing planning? Can't
it be evaluated separately?

- please be consistent and either use IA and IF everywhere or incident arc and
inverted fork factoring everywhere.

- "When enabling pInG, [...], gets the total coverage back" -> "When enabling
pInG, [...], total coverage gets back"

- "don't" -> do not

- runtime vs total time vs "total time" (I don't see why the quotation marks):
this is called total time in Fast Downward/lab, yes, but I think it makes sense
to stick to runtime (or you could be precise and say that it doesn't include
the translator runtime).

- several places: you refer to some plot with technique X (left) and technique
Y, without writing "(right)" (Figures 6, 7, and 8, text and captions)

- Figure 6 mentions "pruning power", but the label of the plot reads
"expansions": please be consistent and use one term

---

> ### Author Response · Authors · 2020-09-07
> **Response to new comments**
>
> Many thanks for the additional feedback and comments, this is very much appreciated! We will consider all comments to further improve the paper.
>
> -- Reg. center/leaf actions: A^C and A^L can indeed overlap. Actions with an effect on (a variable in) C are always center actions and are therefore in A^C. Center actions with an effect on a leaf L are additionally contained in A^L. We will add a note to clarify.
>
> -- Reg. evaluation of INV: both optimizations, for successor generator and dominance checking, have a rather small impact on the performance and are only applicable in few instances. We decided to only textually summarize these findings and focus on the more interesting results. We will give this a second thought for the final version.

---

### Comment · Program_Chairs · 2020-09-14
**Final Decision: Accept**

Dear Authors,

Thank you very much for your submission. We are happy to inform you that we have decided to accept it and we look forward to your talk in the workshop. You will receive additional information per mail in the coming days.

Best,
The HSDIP'20 team

---

### Decision · Program_Chairs · 2020-09-30

Accept